

# A kriging-based analysis of cloud Liquid Water Content using CloudSat data

Jean-Marie Lalande[1], Guillaume Bourmaud[1], Pierre Minvielle[2], and Jean-François Giovannelli[1]

[1]IMS (Univ. Bordeaux, CNRS, Bordeaux INP), 33400 Talence, France
[2]CESTA, DAM, CEA, 33114 Le Barp, France

**Correspondence:** Jean-Marie Lalande (jean-marie.lalande@meteo.fr)

**Abstract.** Spatiotemporal statistical learning has received increased attention in the past decade, due to spatially and temporally indexed data proliferation, especially collected from satellite remote sensing. In the mean time, observational studies of clouds are recognized as an important step to improve cloud representation in weather and climate models. Since 2006, the satellite CloudSat of NASA carries a 94 GHz cloud profiling radar and is able to retrieve, from radar reflectivity, microphysical parameter distribution such as water or ice content. The collected data is piled up with the successive satellite orbits of nearly two hours, leading to a large compressed database of 2 Tb (http://cloudsat.atmos.colostate.edu/).

These observations give the opportunity to extend the cloud microphysical properties beyond the actual measurement locations using an interpolation and prediction algorithm. In order to do so, we introduce a statistical estimator based on the spatiotemporal covariance and mean of the observations known as kriging. An adequate parametric model for the covariance and the mean is chosen from an exploratory data analysis. Beforehand, it is necessary to estimate the parameters of this spatiotemporal model; This is performed in a Bayesian setting. The approach is then applied to a subset of the CloudSat dataset.

## 1 Introduction

Clouds have a strong influence on weather and climate. They are a key element of Earth's hydrological cycle, bringing water from the air to the ground and from one region of the globe to another. They also dominate the energy budget of the Earth through their action on the exchange of solar and thermal radiation within the atmosphere and between the atmosphere, the hydrosphere, the land surface, the biosphere, and space. However, they still remain a major source of uncertainties in predicting the weather and climate change.

While the measurement of cloud occurrences and properties at useful spatial and temporal scales is notoriously difficult (Marshak and Davis, 2005; Stephens and Kummerow, 2007), the proliferation of satellite platforms in the last decades (Stephens et al., 2002; Eriksson et al., 2008; Wu et al., 2009) is fostering a number of new approaches. One such satellite, CloudSat, whose payload is a Cloud Profiling Radar (CPR), has been dedicated to measure the cloud vertical structure and microphys-





ical properties. It is part of the A-train constellation, who was originally set to comprise 7 satellites specifically designed to measure cloud and precipitation properties using different instruments. Since 2006 CloudSat has collected a large database of
cloud properties, globally and over an extended period of time, despite some malfunctions.

In this study, we propose to analyze statistically a part of this database in order to perform interpolation and prediction of cloud properties. It is of major importance for the assessment of satellite cloud attenuation Lyras et al. (2016), including Global Positioning System (GPS) radio occultation Yang and Zou (2012). It is required for improving the representation of cloud systems in numerical weather prediction Bodas-Salcedo et al. (2008); Chen et al. (2011), e.g. for data assimilation Qu
et al. (2018). It can be involved in the assessment of an aircraft icing detection system Vivekanandan et al. (2001), the design of a satellite communication system Khan et al. (2012), the systematic comparison with other cloud products from different instruments/satellites, etc.

As observations from different instruments are likely not collected at the same spatiotemporal positions, the procedure of interpolation-prediction usually involves regridding selected slots of data. We use an approach based on a kriging estimator for
the interpolation/prediction problem. This approach is based on a second-order analysis of cloud Liquid Water Content (LWC) obtained from Level 2B product of the CloudSat ground segment data. We perform a detailed analysis of those properties and propose a parametric model for their mean and covariance. The model parameters are obtained by maximizing the a posteriori probability density which comprises a likelihood term expressing how the model fits the observation conditionnally to the presence of clouds. Finally we apply this model in the context of interpolation and prediction of CloudSat observations.
Kriging estimator have been widely used in the field of geosciences in order to interpolate geophysical properties characterized by spatial or spatiotemporal variability (*i.e.,* mining, hydrogeology, geothermal energy, *etc...*). To our knowledge, it is the first time it is employed to infer cloud microphysical properties using CloudSat observations :

- We provide a comprehensive mathematical description of the kriging estimator. It allows us to mention that the parameters of the mean and covariance functions are commonly treated differently in the kriging literature. In our approach,
parameters can enter both linearly and non-linearly in the mean, which makes it more flexible in accommodating the trend of the observations. Thus, we propose a global treatment of all the parameters of the model as the standard universal kriging is not compatible with non-linearity.

- We propose a model for the mean and covariance functions and thoroughly justify each choice we make (*e.g.,* stationarity assumption, homogeneity assumption, *etc...*)

- We perform a detailed analysis of both the estimated model parameters and the resulting interpolated/predicted cloud LWC.





## 2 Data and Mathematical Description

### 2.1 The CloudSat data and the CPR instrument

CloudSat has been flying in formation in the A-train with other satellites including *Aqua*, *Aura* and CALIPSO. CloudSat
payload consists of a 94 GHz cloud-profiling radar (CPR) that was specifically designed to sense cloud-sized particles (*i.e.,*
cloud ice, snow, cloud droplets and light rain).

It was declared operational on June 2-nd 2006 and has been flying in the A-train until February 22-nd 2018. It follows a Sun-synchronous orbit with an approximately 1330-LT equatorial crossing time. Since 2011 and a battery malfunction, it provides observations only during daytime. The satellite visits the same position of the globe after a period of 16 days corresponding to
233 orbits. Each orbit is achieved in about 1 hour and 58 minutes. The CloudSat radar samples profiles at 625 kHz and has an along-track velocity of approximately 7 km/s, which corresponds to a profile measured every 0.16 second with an along-track displacement of approximately 1.1 km. Each profile has 125 vertical bins of 240 m thickness.

In this study, we use the level 2B Cloud Water Content product (2B-CWC-RO) that contains retrieved estimates of cloud Liquid and Ice Water Content, effective radius, and related quantities for each radar backscattered reflectivity profile (Austin
et al., 2009). We focus on a region centered over Europe and under-sampled at a rate of $1/50$ (*cf.* Fig. 1). In this region, we select data from June 16-th 2006 to June 14-th 2015 constituting a set of 239087 profiles. The CPR observations are classified according to their cloud types in the level 2B-CLDCLASS product (*cf.* Fig. 2). After analysis of the cloud types distribution inside the considered region, we decided to focus on the LWC labeled as Altostratus as they constitute a good balance between the amount of available data and the computational feasibility. Moreover, altostratus are less fractionated than other cloud
types, thereby their physical properties are more continuous.

### 2.2 Mathematical description of the data

#### 2.2.1 Background

We are considering the scalar function $\Phi$ of three spatial variables $(x,y,z)$ and a temporal variable $t$ mapping elements from $D$ into $\mathbb{R}$:

$$
\begin{aligned}
\Phi : \quad D = [0,2\pi] \times [-\pi/2, \pi/2] \times \mathbb{R}^+ \times \mathbb{R} \quad &\rightarrow \quad \mathbb{R} \\
\boldsymbol{s} = (x,y,z,t) \quad &\mapsto \quad \Phi(x,y,z,t),
\end{aligned}
\tag{1}
$$

where $\boldsymbol{s} \in D$ is spatiotemporal localization, $x$, $y$ represent respectively the longitude and latitude at the Earth's surface and $z$ is the altitude. The function $\Phi$ can be modeled as a deterministic or stochastic quantity. The stochastic nature of the function $\Phi$ can be introduced to model some inner variability of the physical phenomenon under study or an incomplete knowledge of the phenomenon itself. The complete description of a stochastic process requires the construction of the joint probability density
function for the continuous variables in space and time (Chonavel and Ormrod, 2002; Gaetan and Guyon, 2008; Brockwell and Davis, 2009), which can be cumbersome in a high-dimensional space. To alleviate this burden, it is possible to resort to the description of the mean and covariance of the random function. This proves very efficient when the distribution associated

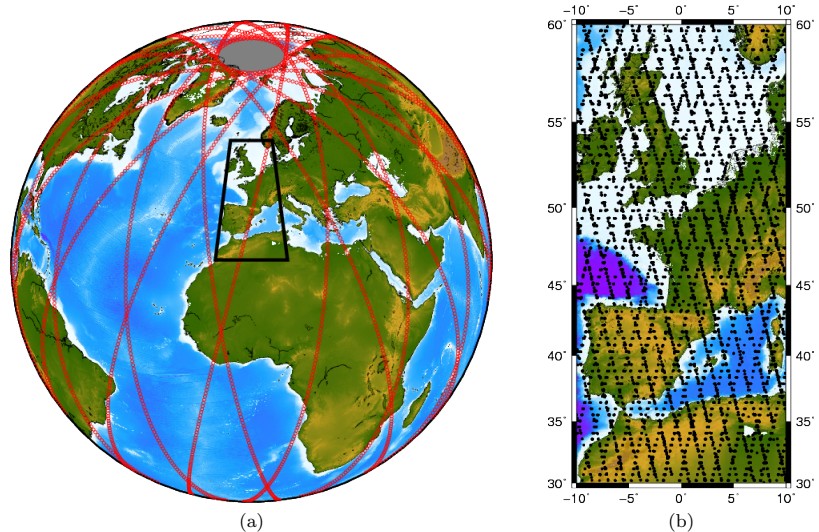

**Figure 1.** CloudSat ground track representing 14 orbits (a), zoom in on the European zone under study (b).

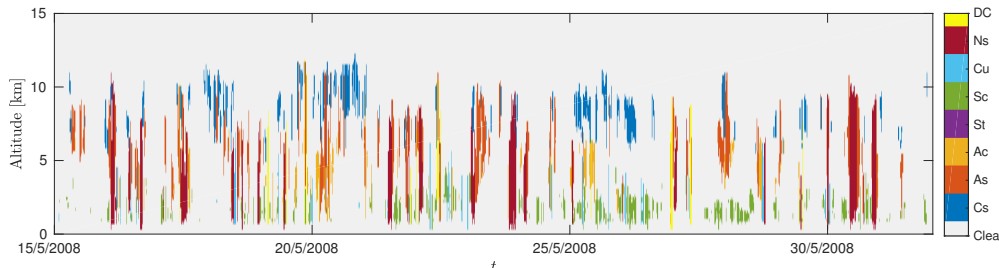

**Figure 2.** Sample of CloudSat cloud classification over a period of 15 days. Clouds are classified according to Clear (no clouds), Cs (Cumulus), As (Altostratus), Ac (Altocumulus), St (Stratus), Sc (Stratocumulus), Cu (Cirrus), Ns (Nimbostratus), DC (Deep Convective).

to the random function is not too complex (for instance when the random function is Gaussian, the mean and covariance are sufficient to completely describe the distribution).

In this study, we assume that the mean and covariance exist and represent sufficiently well the distribution of the random function $\Phi$. The mean is a function of the spatiotemporal localization $\boldsymbol{s}$:

$$m_\Phi(\boldsymbol{s}) = \mathrm{E}\big[\Phi(\boldsymbol{s})\big],$$

where $\mathrm{E}\big[\cdot\big]$ denotes the expectation of the random variable. The covariance is a function of 4 spatiotemporal variables and 4 spatiotemporal shift variables, noted $\boldsymbol{\delta} = (\delta_x, \delta_y, \delta_z, \delta_t)$. Thus we have,

$R_\Phi(\boldsymbol{s}, \boldsymbol{\delta}) = \mathrm{E}\big[(\Phi(\boldsymbol{s}) - m_\Phi(\boldsymbol{s}))(\Phi(\boldsymbol{s} + \boldsymbol{\delta}) - m_\Phi(\boldsymbol{s} + \boldsymbol{\delta}))\big].$





The covariance function describes the spatiotemporal dependency of the stochastic function $\Phi$. Based on these definitions, it is useful to introduce certain stationary model to characterize the degree of homogeneity of the random function $\Phi$.

### 2.2.2 Stationarity

In order to infer the distribution of the random function $\Phi$, the structure of the spatiotemporal process is of great importance. In this sense, stationary model enable to structure the spatiotemporal variability of the random function $\Phi$. The process under study is said to be strictly stationary if the distributions $[\Phi(\boldsymbol{s}_1),\cdots,\Phi(\boldsymbol{s}_k)]$ and $[\Phi(\boldsymbol{s}_1+\boldsymbol{\delta}),\cdots,\Phi(\boldsymbol{s}_k+\boldsymbol{\delta})]$ are identical for all $\boldsymbol{\delta}, \boldsymbol{s}_1,\ldots,\boldsymbol{s}_k \in D$, and for all $k \in \mathbb{N}$. This is a very restrictive condition which is not usually satisfied in real-life applications. The weak stationarity stipulates that:

- the random function $\Phi$ is 1-st order stationary when its expected value does not depend on the localization $\boldsymbol{s}$:

$$m_\Phi(\boldsymbol{s}) = \mu$$

- the random function $\Phi$ is 2-nd order stationary when its covariance function depends only on the lag vector $\boldsymbol{\delta}$ between 2 localizations $\boldsymbol{s}$ and $\boldsymbol{s}' = \boldsymbol{s} + \boldsymbol{\delta}$:

$$R_\Phi(\boldsymbol{s},\boldsymbol{\delta}) = R_\Phi(\boldsymbol{\delta}), \text{for all } \boldsymbol{s} \text{ and } \boldsymbol{\delta}.$$

### 2.2.3 Observations and uncertainties

We denote the successive spatiotemporal positions $\boldsymbol{s}_n = (x_n,y_n,z_n,t_n)$, $n = 1,\cdots,N$ and the targeted quantity $\Phi$ at these positions:

$$\Phi_n = \Phi(\boldsymbol{s}_n), \quad n = 1,\cdots,N.$$

We model the actual observations of the targeted quantity $\Phi$ as corrupted by an additive noise related to the measurement uncertainty. The observation at position $\boldsymbol{s}_n$ is then denoted $\Psi_n$ and is written:

$$\Psi_n = \Phi_n + B_n,$$

where $\Psi_n$, $\Phi_n$ and $B_n$ are random variables. We denote the values associated with a realization of these random variables by $\psi_n$, $\varphi_n$ and $b_n$. Thus, a realization of the random variable $\Psi$ at the position $\boldsymbol{s}_n$ is written:

$$\psi_n = \varphi_n + b_n, \tag{2}$$

and additionally, we write:

$$\boldsymbol{\psi} = \boldsymbol{\varphi} + \boldsymbol{b},$$

where $\boldsymbol{\psi} = [\psi_1,\cdots,\psi_N]$, $\boldsymbol{\varphi} = [\varphi_1,\cdots,\varphi_N]$ and $\boldsymbol{b} = [b_1,\cdots,b_N]$ are the collections of the related $N$ quantities.





### 2.2.4 Objectives

The objective of this work is to determine the value $\varphi_0$ of the quantity of interest $\varphi$ at a given location $\boldsymbol{s}_0 = (x_0, y_0, z_0, t_0)$. This
is an estimation problem that is tackled by designing an estimate $\hat{\varphi}_0$ of the quantity of interest $\varphi_0$ from the data $\boldsymbol{\psi}$ described
in the previous section. On the one hand, spatially, this is an interpolation problem since the point $x_0, y_0, z_0$ is usually not
on the grid of observations. On the other hand, temporally, this is a prediction problem since $t_0$ is naturally positioned in the
future. The considered approach is to statistically learn from the large amount of available data both to interpolate/predict the
targeted quantity and to quantize the uncertainty associated with the estimated value. In the next section, we introduce the
kriging estimator, that is specifically designed to perform such a task.

## 3   The kriging estimator

The goal is to construct a statistical estimator $\hat{\Phi}_0$

$$
\begin{aligned}
\hat{\Phi}_0 : \quad \mathbb{R}^N &\rightarrow \mathbb{R} \\
\boldsymbol{\psi} &\mapsto \hat{\varphi}_0 = \hat{\Phi}_0(\boldsymbol{\psi})
\end{aligned}
$$

in order to estimate $\varphi_0$ from the observations $\boldsymbol{\psi}$. This section describes the strategy for the construction of the kriging estimator
which has been widely used in geostatistics. This framework has been developed in a handful of handbooks (Cressie, 1993;
Chiles and Delfiner, 1999; Diggle et al., 2003) and later on by (Montero et al., 2015), in the case of a known mean and
covariance. Our development differs slightly from the standards of geostatistical literature in two ways.

1. The parameters of the mean function $m_\Phi(\boldsymbol{s})$ usually act linearly and consequently their estimation can be performed
inside the kriging estimator while the estimation of the covariance parameters is classically performed in a previous
offline procedure. Instead, in order to design a model with a higher capacity, we consider both a non-linear mean and a
non-linear covariance functions and estimate their parameters in an offline procedure.

2. Our approach is fully parametric and consequently relies on a parametric covariance function instead of a variogram
(Banerjee et al., 2014).

Thus, in the following development, we consider that the mean and covariance functions are known and defer the (offline)
estimation of their parameters to the next section.
We consider a linear estimator of the form:

$$
\hat{\Phi}_0(\boldsymbol{\Psi}) = \boldsymbol{a}^{\mathrm{t}} \boldsymbol{\Psi} + a_0, \tag{3}
$$

where $\boldsymbol{\Psi} = [\Psi_1, \cdots, \Psi_N]$ are the observations, $\boldsymbol{a} = [a_1, \cdots, a_N]$ are scalar coefficients and $N$ is the number of available obser-
vations. This can be alternatively expressed in terms of realizations of the random variables as:

$$
\hat{\varphi}_0 = \hat{\Phi}_0(\boldsymbol{\psi}) = \boldsymbol{a}^{\mathrm{t}} \boldsymbol{\psi} + a_0. \tag{4}
$$





Our goal is to determine the coefficients $\boldsymbol{a}$ and $a_0$. The strategy is to minimize an error between the estimated quantity $\hat{\Phi}_0$ and the true value $\Phi_0$. We choose the Mean Squared Error (MSE),

$$\mathcal{E}(\boldsymbol{a}, a_0) = \mathrm{E}\left[(\hat{\Phi}_0 - \Phi_0)^2\right]. \tag{5}$$

The estimator $\hat{\Phi}$, defined by (3)-(4) with:

$$(\boldsymbol{a}, a_0)^{\mathrm{opt}} = \underset{\boldsymbol{a}, a_0}{\arg\min} \; \mathcal{E}(\boldsymbol{a}, a_0),$$

and the corresponding estimator is the so-called Minimum Mean Square Error estimator (MMSE). The value $a_0^{\mathrm{opt}}$ minimizing (5) satisfies

$$\frac{\partial \mathcal{E}}{\partial a_0}\bigg|_{a_0^{\mathrm{opt}}} = 0$$

giving,

$$a_0^{\mathrm{opt}} = m_{\Phi_0} - \boldsymbol{a}^{\mathrm{t}} \boldsymbol{m}_{\Phi}. \tag{6}$$

Then, plugin (6) into (5) and expanding leads to:

$$\mathcal{E}(\boldsymbol{a}) = \mathcal{E}(\boldsymbol{a}, a_0^{\mathrm{opt}}) = \boldsymbol{a}^{\mathrm{t}} \boldsymbol{R_{\Psi}} \boldsymbol{a} - 2\boldsymbol{a}^{\mathrm{t}} \boldsymbol{r}_{\boldsymbol{\Psi}\Phi_0} + \mathrm{var}\left[\Phi_0\right] \tag{7}$$

where we introduced the following notations:

$$\boldsymbol{R_{\Psi}} = \mathrm{E}\left[(\boldsymbol{\Psi} - \boldsymbol{m_{\Phi}})(\boldsymbol{\Psi} - \boldsymbol{m_{\Phi}})^{\mathrm{t}}\right]$$
$$\boldsymbol{r}_{\boldsymbol{\Psi}\Phi_0} = \mathrm{E}\left[(\Phi_0 - m_{\Phi_0})(\boldsymbol{\Psi} - \boldsymbol{m_{\Phi}})\right]$$
$$\mathrm{var}\left[\Phi_0\right] = \mathrm{E}\left[(\Phi_0 - m_{\Phi_0})^2\right],$$

$\boldsymbol{R_{\Psi}}$ is the covariance matrix of the observations, $\boldsymbol{r}_{\boldsymbol{\Psi}\Phi_0}$ the covariance between the observations and the quantity of interest, and $\mathrm{var}\left[\Phi_0\right]$ is the variance of the quantity of interest. The vector $\boldsymbol{a}^{\mathrm{opt}}$ that minimizes (7) satisfies

$$\frac{\partial \mathcal{E}}{\partial \boldsymbol{a}}\bigg|_{\boldsymbol{a}^{\mathrm{opt}}} = \boldsymbol{0},$$

and that leads to:

$$\boldsymbol{a}^{\mathrm{opt}} = \boldsymbol{R_{\Psi}}^{-1} \boldsymbol{r}_{\boldsymbol{\Psi}\Phi_0}.$$

Moreover, modeling noise $\boldsymbol{B}$ independent of the quantity of interest $\Phi$, we have

$$\boldsymbol{R_{\Psi}} = \boldsymbol{R_{\Phi}} + \boldsymbol{R_B} \quad \text{and} \quad \boldsymbol{r}_{\boldsymbol{\Psi}\Phi_0} = \boldsymbol{r}_{\boldsymbol{\Phi}\Phi_0}, \tag{8}$$

so that,

$$\boldsymbol{a}^{\mathrm{opt}} = (\boldsymbol{R_{\Phi}} + \boldsymbol{R_B})^{-1} \boldsymbol{r}_{\boldsymbol{\Phi}\Phi_0}. \tag{9}$$



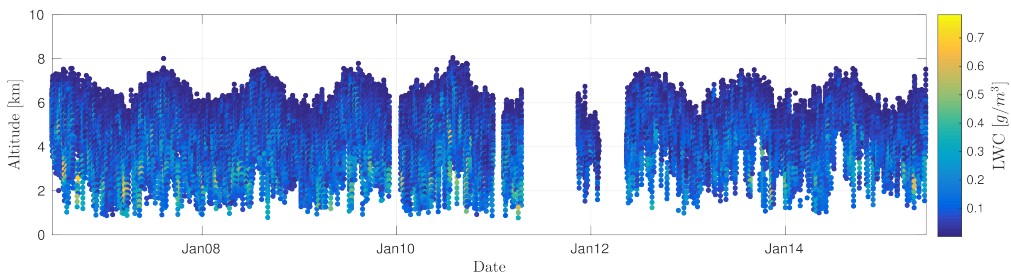

**Figure 3.** Scatter plot of the altostratus LWC observations.

Finally, we deduce the estimated value $\hat{\varphi}_0$ of the quantity of interest $\varphi$ at position $\boldsymbol{s}_0$ by inserting equations (9) and (6) in (4):

$$\hat{\varphi}_0 = \boldsymbol{r}_{\boldsymbol{\Phi}\Phi_0}^{\mathrm{t}} \left(\boldsymbol{R}_{\boldsymbol{\Phi}} + \boldsymbol{R}_B\right)^{-1} \left(\boldsymbol{\psi} - \boldsymbol{m}_{\boldsymbol{\Phi}}\right) + m_{\boldsymbol{\Phi}}(\boldsymbol{s}_0). \tag{10}$$

Additionally, the MSE of the estimated value is given by:

$$\mathcal{E}(\boldsymbol{a}^{\mathrm{opt}}, \boldsymbol{a}_0^{\mathrm{opt}}) = \mathrm{var}\left[\Phi_0\right] - \boldsymbol{r}_{\boldsymbol{\Phi}\Phi_0}^{\mathrm{t}} \left(\boldsymbol{R}_{\boldsymbol{\Phi}} + \boldsymbol{R}_B\right)^{-1} \boldsymbol{r}_{\boldsymbol{\Phi}\Phi_0}, \tag{11}$$

which, in this context, equals to the variance of the estimation error $\mathrm{var}\left[\hat{\Phi}_0 - \Phi_0\right]$ because the bias is equal to zero. This estimator is known as the kriging estimator in the geostatistical literature (Montero et al., 2015). In a slightly different form, this estimator is also the one of Kalman and Wiener in the field of filtering.

# 4 Construction of a model for the mean and the covariance

## 4.1 Time-altitude and spectral analysis

In order to construct an adequate model for the mean and the covariance, we start with an exploratory analysis to extract some raw characteristics from the time-altitude observations represented in Fig. 3. We compute the biased empirical covariance for the 2-dimensional case $(\delta t, \delta z)$ and the 1-dimensional case at a fixed altitude of $z = 6.711$ km (*cf.* Fig. 4). This shows 1 year periodicity of the LWC observations in the time component, while the altitude component decreases to 0 for $\delta z = 1$ km. The blue spots located at $\delta_z = +/-6$ km are due to zero-padding above and below the cloudy profile that introduces an artificial

correlation. Another way of looking at the data $\boldsymbol{\psi}$ is to compute the empirical power spectral density $\hat{S}_{\boldsymbol{\psi}}(f)$, *i.e.,* the squared modulus of the Fourier transform of the observations $\boldsymbol{\psi}$ (a.k.a. periodogram) (*cf.* Fig. 5). There is clearly a peak at $1$ year$^{-1}$. For some altitudes, there are lower peaks at $2$ year$^{-1}$ or even at $3$ year$^{-1}$, indicating that the signal cannot be modeled by a single sinusoidal component. At lower altitudes, the fundamental component becomes negligible as well as the harmonics. Nevertheless, this suggests that CloudSat observations can be interpreted as the superposition of multiple periodic components.





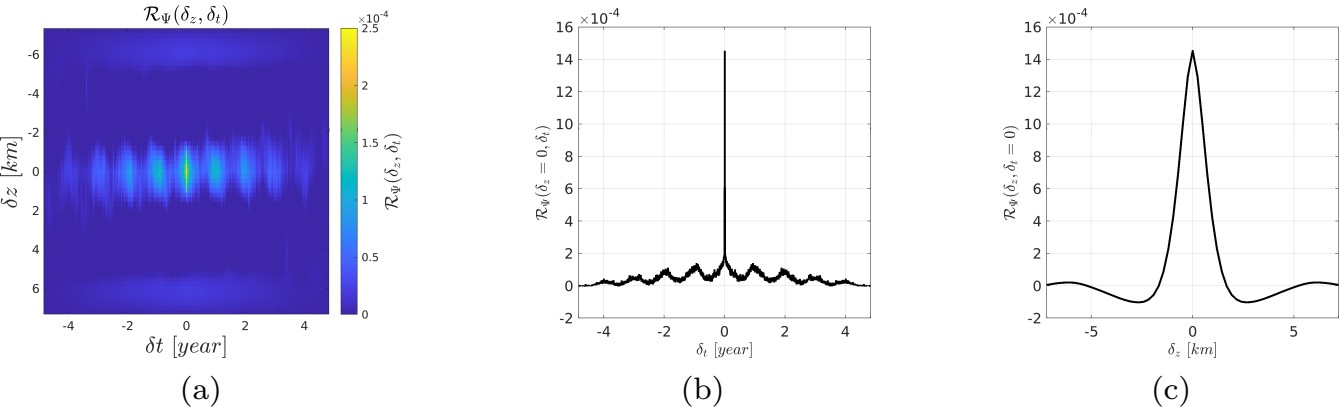

**Figure 4.** Empirical covariance: (a) 2-dimensional covariance $(\delta t, \delta z)$, (b) slice at $\delta_z = 0$ km and (c) slice at $\delta_t = 0$ year.

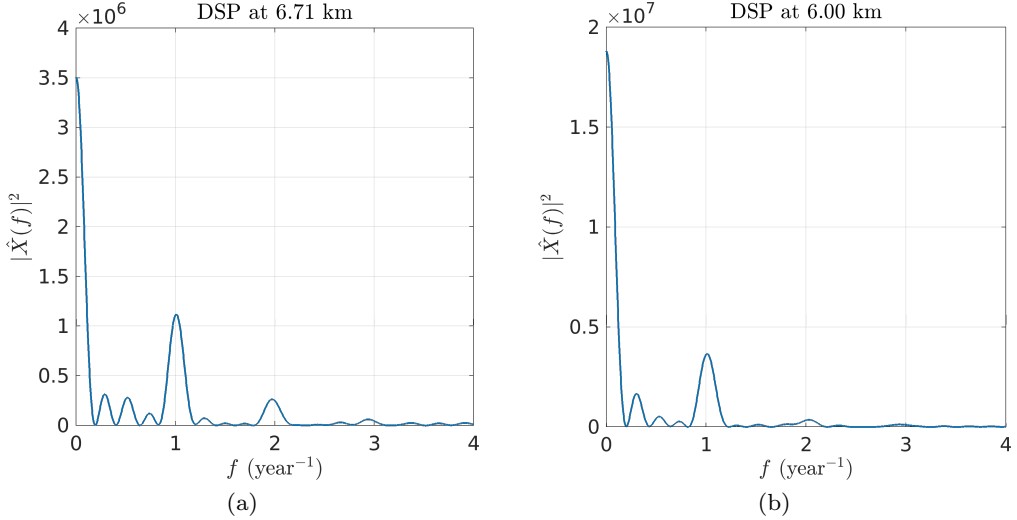

**Figure 5.** Empirical periodogram at (a) $z = 6.71$ km and (b) $z = 6.00$ km.

## 4.2 Proposition of an adequate model

The observations $\Psi$ are modeled as the addition of a random process $\Phi$ and a random noise process $B$, such as:

$$\Psi(t, z) = \Phi(t, z) + B(t, z).$$

We dodge the complexity of handling 4 dimensions and choose to do an in-depth analysis of a time-altitude dependent model while omitting the latitude-longitude dimensions. This simplification can be justified when the spatial extension of the dataset is not too important. This is equivalent to assimilate the dataset to a single spatial location by averaging over the geographic





components. All random processes are modeled as Gaussian (Rasmussen and Williams, 2005):

$$\Phi \sim \mathcal{N}(m_\Phi, \boldsymbol{R}_\Phi),$$

$$B \sim \mathcal{N}(\boldsymbol{0}, \boldsymbol{R}_B),$$

$$\Psi \sim \mathcal{N}(m_\Phi, \boldsymbol{R}_\Psi), \tag{12}$$

where $\boldsymbol{R}_\Phi$, $\boldsymbol{R}_B$ and $\boldsymbol{R}_\Psi$ are the covariance matrices associated to the quantity of interest $\Phi$, the noise $B$ and the observations $\Psi$. Assuming the noise $B$ independent of the random function $\Phi$, we can write: $\boldsymbol{R}_\Psi = \boldsymbol{R}_\Phi + \boldsymbol{R}_B$, see (8).

In order to take into account the periodic temporal trend observed in the observations, we consider a non 1-st order stationary model. Moreover, the mean is chosen not to be a function of the altitude in order to limit the number of parameters. Considering the time variations, we chose a periodic mean with 1 fundamental and 2 harmonics such that:

$$m_\Phi(t) = \beta_0 + \beta_1 \cos(\alpha_1 + \frac{2\pi t}{T})$$
$$+ \beta_2 \cos(\alpha_2 + \frac{2\pi t}{T/2}) + \beta_3 \cos(\alpha_3 + \frac{2\pi t}{T/3})$$

where $\beta_0$, $\beta_1$, $\beta_2$ and $\beta_3$ are respectively the amplitude of the continuous component, the fundamental and the 2 harmonics, $\alpha_i$
(with $i = 1, 2, 3$) are the phase parameters of the fundamental and the harmonics and $T$ is the period.

In order to further simplify we resort to the widely used separable model for the covariance (Genton, 2007) so that the spatio-temporal covariance factors into a purely spatial and a purely temporal component. This reduces the number of parameters allowing for computationally efficient estimation and inference, it is then more tractable to run different test on the dataset. Thus, the separable covariance model writes

$$\mathcal{R}_\Phi(\delta_t, \delta_z) = r_\Phi \, \mathcal{R}_{\Phi_t}(\delta t) \, \mathcal{R}_{\Phi_z}(\delta z),$$

where $r_\Phi$ is the variance of the quantity of interest. We chose an exponential covariance model for the time covariance

$$\mathcal{R}_{\Phi_t}(\delta t) = \exp[-|\delta t|/l_t],$$

where $l_t$ is the correlation time. The choice of this covariance model is motivated by the shape of the empirical covariance described in the previous section (it decreases more rapidly than a Gaussian). The altitude covariance is set to be:

$$\mathcal{R}_{\Phi_z}(\delta z) = \exp\left[-(\delta z/l_z)^2\right],$$

where $l_z$ is the correlation length in altitude. Finally, we use a white noise:

$$\mathcal{R}_B(\delta t, \delta z) = r_B \, \mathrm{Dirac}(\delta t, \delta z),$$

where $r_B$ is the variance of the noise. Finally, we have a model consisting of 12 parameters, including 4 parameters for covariance, and 8 parameters for the mean

$$\boldsymbol{\theta} = [r_\Phi, l_t, l_z, \beta_0, \beta_1, \beta_2, \beta_3, \alpha_1, \alpha_2, \alpha_3, T, r_B].$$

Using this notation we can denote the parametric model for the mean $\boldsymbol{m}_\Psi(\delta t; \boldsymbol{\theta})$ and for the covariance $\boldsymbol{R}_\Psi(\delta t, \delta z; \boldsymbol{\theta})$. In the following section, we describe the strategy to estimate this set of parameters.



## 5 Model parameter estimation, optimization and final kriging equations

In this section, we present the strategy for the estimation of the model parameters $\boldsymbol{\theta}$. We choose the Maximum A Posteriori
(hereafter abbreviated MAP) estimate for $\boldsymbol{\theta}$ which is the mode of the *posterior* probability density function (pdf) in Bayesian
statistics. The analysis of the estimated parameters and corresponding model is deferred to Sect. 6 for the 1-dimensional case
and Sect. 7 for the 2-dimensional case.

### 5.1 The MAP estimator

In the previous section, we proposed a parametric model for the mean $\boldsymbol{m}_\Psi(\delta t; \boldsymbol{\theta})$ and for the covariance $\boldsymbol{R}_\Psi(\delta t, \delta z; \boldsymbol{\theta})$ of the
random variables $\Psi$, which decomposes into a model for the quantity of interest $\Phi$ and the noise $B$. The problem is to find an
estimate $\hat{\boldsymbol{\theta}}$ of the true parameters $\boldsymbol{\theta}^\star$ from the observations $\boldsymbol{\psi}$. We introduce the *posterior* pdf:

$$\pi(\boldsymbol{\theta}|\boldsymbol{\psi}) = \frac{f(\boldsymbol{\psi}|\boldsymbol{\theta})\rho(\boldsymbol{\theta})}{f(\boldsymbol{\psi})},$$

where $f(\boldsymbol{\psi}|\boldsymbol{\theta})$ is the so-called likelihood function, which expresses the probability of the observations given the parameter $\boldsymbol{\theta}$,
$\rho(\boldsymbol{\theta})$ is the *prior* distribution for the parameters $\boldsymbol{\theta}$ and $f(\boldsymbol{\psi})$ is the marginal pdf for the observations. Since we model the
observations by a Gaussian pdf conditionally to the parameters $\boldsymbol{\theta}$ (see (12)), the likelihood function $f(\boldsymbol{\psi}|\boldsymbol{\theta})$ writes:

$$f(\boldsymbol{\psi}|\boldsymbol{\theta}) = (2\pi)^{-N/2}(\det \boldsymbol{R}_\Psi)^{-1/2}$$
$$\exp\left[-\frac{1}{2}(\boldsymbol{\psi}-\boldsymbol{m}_\Psi)^{\mathrm{t}}\boldsymbol{R}_\Psi^{-1}(\boldsymbol{\psi}-\boldsymbol{m}_\Psi)\right], \tag{13}$$

where we have omitted the dependence in $\boldsymbol{\theta}$, $\delta t$ and $\delta z$ for $\boldsymbol{R}_\Psi$ and $\boldsymbol{m}_\Psi$ to weigh down the notations. Since the *posterior* pdf
$\pi(\boldsymbol{\theta}|\boldsymbol{\psi})$ is by definition positive, we consider the co-log posterior $\mathrm{CLP}(\boldsymbol{\theta}) = -\log\pi(\boldsymbol{\theta}|\boldsymbol{\psi})$, which expands as follows:

$$\mathrm{CLP}(\boldsymbol{\theta}) \,\#\, -\log f(\boldsymbol{\psi}|\boldsymbol{\theta}) - \log\rho(\boldsymbol{\theta})$$
$$\#\, \frac{1}{2}(\boldsymbol{\psi}-\boldsymbol{m}_\Psi)^{\mathrm{t}}\boldsymbol{R}_\Psi^{-1}(\boldsymbol{\psi}-\boldsymbol{m}_\Psi)$$
$$+\frac{1}{2}\log\det\boldsymbol{R}_\Psi - \log\rho(\boldsymbol{\theta}) \tag{14}$$

where $\#$ stands for "equality up to an additive constant" (not function of $\boldsymbol{\theta}$). The minimum of (14) is the maximum of the
*posterior* pdf. Thus, minimizing the CLP gives the MAP estimate. To complete, we consider a uniform *prior* $\rho(\boldsymbol{\theta})$ on a given
domain $\Theta$:

$$\rho(\boldsymbol{\theta}) = \mathcal{U}_\Theta(\boldsymbol{\theta})$$

where $\Theta$ is a hyper-rectangle in $\mathbb{R}^P$ describing a range of possible values for each component of $\boldsymbol{\theta}$. For our model $P = 12$.
Finally, we obtain:

$$\mathrm{CLP}(\boldsymbol{\theta}) \,\#\, (\boldsymbol{\psi}-\boldsymbol{m}_\Psi)^{\mathrm{t}}\boldsymbol{R}_\Psi^{-1}(\boldsymbol{\psi}-\boldsymbol{m}_\Psi)$$
$$+\log\det\boldsymbol{R}_\Psi - \log\mathcal{U}_\Theta(\boldsymbol{\theta}) \tag{15}$$





The function CLP is infinite when $\boldsymbol{\theta} \notin \boldsymbol{\Theta}$ because of the term $-\log \mathcal{U}_{\boldsymbol{\Theta}}(\boldsymbol{\theta})$. The minimizer $\hat{\boldsymbol{\theta}}$, given by:

$$\hat{\boldsymbol{\theta}} = \underset{\boldsymbol{\theta}}{\arg\min} \ \mathrm{CLP}(\boldsymbol{\theta}), \tag{16}$$

is the MAP estimate of the true parameter $\boldsymbol{\theta}^{\star}$.

## 5.2 Optimization procedure

An optimization procedure is required in order to determine the minimizer $\hat{\boldsymbol{\theta}}$ of the CLP. In this study, we use an Iterative Conditional Mode strategy with a Golden-Section search (ICM-GS) for each component (Kiefer, 1953; Press et al., 1992). We then have an algorithm composed of an outer and an inner loop.

The inner loop optimizes w.r.t. one component at a time, say $\theta_p$, based on a golden-section search. It stops when the variation of $\theta_p$ is smaller than a given threshold, say $\varepsilon_p$. The outer loop repeats the scan of the $P$ components and stops when the variation

of $\boldsymbol{\theta}$ becomes smaller than a second given threshold denotes by $\eta$. The values $\varepsilon_p$ and $\eta$ have been fixed so as to reach stable results. The specific threshold values used in optimization procedure are reported in Table 1.

**Table 1.** Threshold values $\epsilon_p$ used for each parameters and global threshold value $\eta$.

| $\epsilon_{r_\Phi}$ | $\epsilon_{l_t}$ | $\epsilon_{l_z}$ | $\epsilon_{r_B}$ | $\epsilon_{\beta_0}$ | $\epsilon_{\beta_1}$ | $\epsilon_{\beta_2}$ | $\epsilon_{\beta_3}$ | $\epsilon_{\alpha_1}$ | $\epsilon_{\alpha_2}$ | $\epsilon_{\alpha_3}$ | $\eta$ |
|---|---|---|---|---|---|---|---|---|---|---|---|
| $1e-5$ | $1e-7$ | $1e-1$ | $1e-6$ | $1e-3$ | $1e-2$ | $1e-2$ | $1e-4$ | $1e-2$ | $1e-5$ | $1e-2$ | $1e-3$ |

## 5.3 Kriging equation with $\hat{m}_\Phi$ and $\hat{R}_\Psi$

The interpolation/prediction solution given the observations $\boldsymbol{\psi}$ is performed using the kriging estimator developed in Sect. 3. The kriging equations were initially developed assuming a known mean $m_\Phi$ and covariance $\boldsymbol{R}_\Psi$. In reality, we only have

at hand the parametric mean $\hat{m}_\Phi$ and covariance $\hat{\boldsymbol{R}}_\Psi$ that depend on the estimated parameter $\hat{\boldsymbol{\theta}}$. Thus, the estimation of the quantity of interest $\Phi$ in $t_0$ is written:

$$\hat{\varphi}_0 = \hat{\boldsymbol{r}}_{\boldsymbol{\Psi}\Phi_0}^{\mathrm{t}} \hat{\boldsymbol{R}}_{\boldsymbol{\Psi}}^{-1} (\boldsymbol{\psi} - \hat{\boldsymbol{m}}_\Phi) + \hat{m}_\Phi(t_0), \tag{17}$$

where $\hat{\boldsymbol{m}}_\Phi$ is a vector of estimated mean at the positions of the observations. Finally, we can compute the minimum mean square error:

$$\mathcal{E} = \hat{r}_\Phi - \hat{\boldsymbol{r}}_{\boldsymbol{\Psi}\Phi_0}^{\mathrm{t}} \hat{\boldsymbol{R}}_{\boldsymbol{\Psi}}^{-1} \hat{\boldsymbol{r}}_{\boldsymbol{\Psi}\Phi_0}. \tag{18}$$

This last expression is equal to the variance of the estimation error, which we note $\sigma^2 = \mathrm{var}\left[\hat{\Phi}_0 - \Phi_0\right]$. It gives an estimation of the forecast error associated with the kriging technique. However, it should be kept in mind that this variance does not account for the uncertainty in the model parameter estimate $\hat{\boldsymbol{\theta}}$. Therefore, this expression tends to underestimate the true estimation variance. This feature has been well documented in (Cressie, 1993; Montero et al., 2015). We leave the assessment of this

impact for further development of this model. Note that the kriging stage is essentially a computational step that does not present any major difficulty except for the computational burden when dealing with large dimensions.





## 6 The 1-dimensional time case

We start our analysis with the 1-dimensional case at a single altitude. We give a detailed analysis of the estimated parameters according to the MAP criterion and the ICM-GS algorithm, both described in the previous section. We concentrate on the
estimation over the west of Europe and analyse the impact of reducing the considered geographic area. We then proceed to the kriging of observations in the interpolation and prediction case.

### 6.1 Estimation over a European area at a single altitude

In order to analyze the results of the previously described parameter estimation procedure, we first consider observations associated to altostratus clouds over an European area extending from $10^{o}$W to $10^{o}$E in longitude and $30^{o}$N to $60^{o}$N in latitude
(see Fig. 1) and use a subsampled database (at $1/50$ rate) at a single altitude of $z \simeq 6$ km. This corresponds to a set of $N_{\mathrm{obs}} = 3653$ observations. The period parameter $T$ has been readily set to 1 year so as to represent the seasonality observed in the data.

Fig. 6 shows the evolution of CLP during the optimization process which, as expected, decreases with the iterations and stabilizes after $\sim 5$ iterations. The current value of the parameters changes at each iteration and stabilizes as shown in Fig. 7.

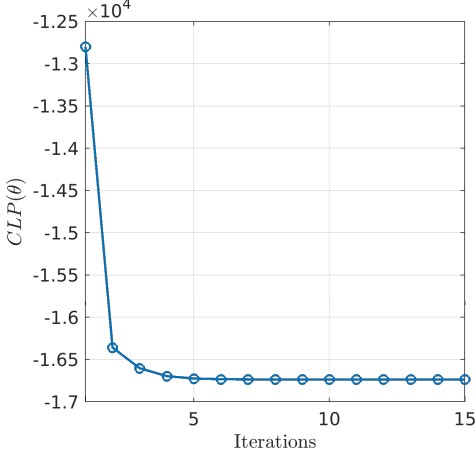

**Figure 6.** Evolution of CLP during the optimization process.

Even though the decrease of CLP along the iterative optimization process indicates convergence towards a minimum, there is no guarantee that it is a global minimum. This is highlighted in Fig. 8(a) which represents the isocontour plot of CLP as a function of $(r_{\Phi}, l_t)$ while the other parameters are fixed to their estimated values. Additionnally, we plot the current values of $r_{\Phi}$ and $l_t$ at each iteration of the ICM-GS algorithm in Fig. 8(a). In Fig. 8(b) we plotted the CLP as a function of $r_{\Phi}$ while fixing all the other parameters to their initial values to highlight the fact that the first two iterations of the optimization algorithm are
moving away from the global minimum shown in the contour plot. The contour plot has been represented by varying values of $r_{\Phi}$ and $l_t$ while setting the other parameters to their estimated value at the end of the optimization procedure so that it is not





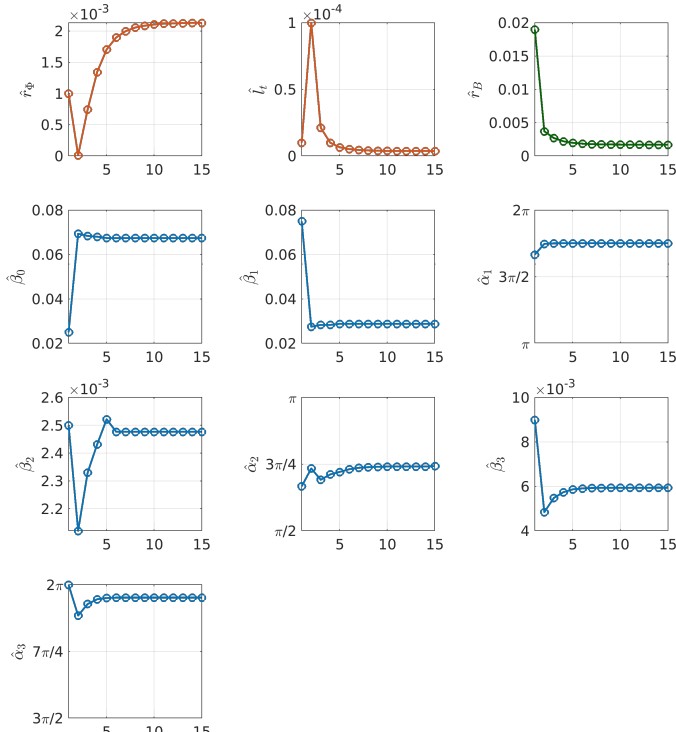

**Figure 7.** Evolution of each parameters along the optimization process. The $x$-axes represents the iteration index while the $y$-axes represent the current value for each parameter. Parameters associated to the covariance of the quantity of interest are represented in red ($\circ$); parameters associated to the mean of the quantity of interest are represented in blue ($\circ$) and the noise variance is represented in green ($\circ$).

representative of the CLP at the initial states. It clearly shows a global and a local minimum. The global minimum is reached for $(\hat{r}_\Phi, \hat{l}_t) \simeq (2 \times 10^{-3}, 3.75 \times 10^{-6})$. The estimated value $\hat{l}_t = 3.75 \times 10^{-6}$ corresponds to $\sim 2$ minutes. This correlation time is very small compared to the 9-year period used to train the parameters. It indicates that there is essentially no correlation

for observations later than 6 minutes apart. Indeed, the exponential covariance model reaches a $5\%$ correlation at about $3l_t$ (Banerjee et al., 2014). This feature can be either explained by the model, the observations or the phenomenon itself. However, it is then evident that such a model will not be very efficient in the perspective of long-term forecast.

As stated in (Rasmussen and Williams, 2005), each local minimum corresponds to a particular interpretation of the data, and a careful analysis may be required to choose one model instead of the other. In the next section, we will further discuss the

presence of two minima of the CLP in relation with two competing models. An exhaustive search for all local minima requires to compute the criterion $\mathrm{CLP}(\boldsymbol{\theta})$ for all possible combinations of parameters in the domain $\boldsymbol{\Theta}$, which is impossible. Instead, we performed a set of 10 optimizations with random initialization to track down additional local minima and converged toward the same minimizer. Hence we conclude there is no other local minima than the one previously pointed out (*cf.* Fig. 8(a)).



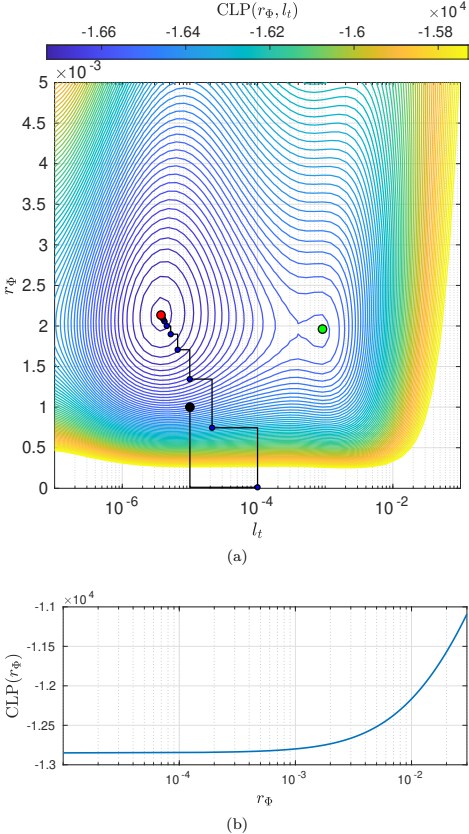

**Figure 8.** (a) Contour plot of CLP as a function of $r_\Phi$ and $l_t$. There is a global minima characterized by $\hat{r}_\Phi \simeq 2.2 \times 10^{-2}$, $\hat{l}_t \simeq 3.6 \times 10^{-6}$ and a corresponding $\mathrm{CLP}(\hat{\theta}) = -1.67 \times 10^4$ and a local minima characterized by $\hat{r}_\Phi \simeq 2 \times 10^{-2}$, $\hat{l}_t \simeq 9.1 \times 10^{-4}$ and $\mathrm{CLP}(\hat{\theta}) = -1.65 \times 10^4$. (b) CLP as a function of $r_\Phi$ while fixing all the other parameters to their initial values. This shows why the first two iterations of the optimization algorithm are moving away from the global minimum shown in the contour plot because the CLP is strictly monotonic inside the considered region.

## 6.2 Discussion on the reduction of the geographic area

In order to assess the influence of the considered geographic area, the optimization procedure is run by reducing the geographic extent of the dataset. We consider 9 areas that are represented in Fig. 9, and the estimated parameters $\hat{\theta}$ are summarized in Fig. 10.

The estimated values of the covariance parameters (*i.e.,* $\hat{r}_\Phi$, $\hat{l}_t$ and $\hat{r}_B$) vary in a complementary fashion, with approximately similar values for zones 1 to 3, a leap of several orders of magnitude for zones 4 to 8, followed by a drop for the 9-th zone. These variations can be traced back to the presence of a local minimum in the domain $\Theta$ as it is shown in Fig. 11 where the variation of $\mathrm{CLP} - \mu_{\mathrm{CLP}}$ with respect to $l_t$ has been represented for each geographic area. All CLPs show the presence of a local and global minima on the considered interval (except for the 9-th area). The variations of CLP w.r.t. the correlation time





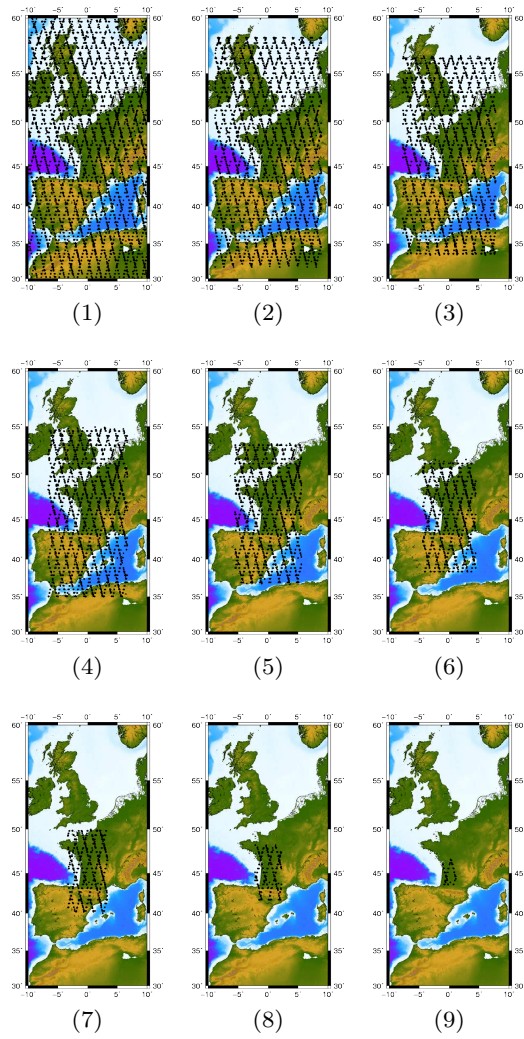

**Figure 9.** The 9 geographic areas used for the parameter estimation analysis.

$l_t$ have similar behavior for areas 1 to 3 with locations of the local and global minimum roughly identical. The same is valid for areas 4 and 5. Concerning areas 6 to 8 (*cf.* Fig. 11(c)), they have a more complex shape with a well defined global minimum

but less pronounced local minima at lesser correlation time.

It is also worth highlighting the sliding of the global minimum to the local minimum when switching from zones 1, 2 and 3 to zones 4 and 5 (*cf.* Fig. 11 (a) and (b)). The global minimum of zones 1, 2 and 3 is reached for a smaller correlation time $l_t \sim 10^{-6}$ year which means that the aggregation of spatially distant observations is interpreted as a less correlated process than for smaller geographic area (*i.e.,* zones 4 and 5). In this case, the retrieved model tends towards a white Gaussian noise

as the covariance function is close to a Dirac. However, the presence of a local minimum of CLP with a higher correlation time indicates that the observations could also be explained by a second model, provided some additional constraint on the



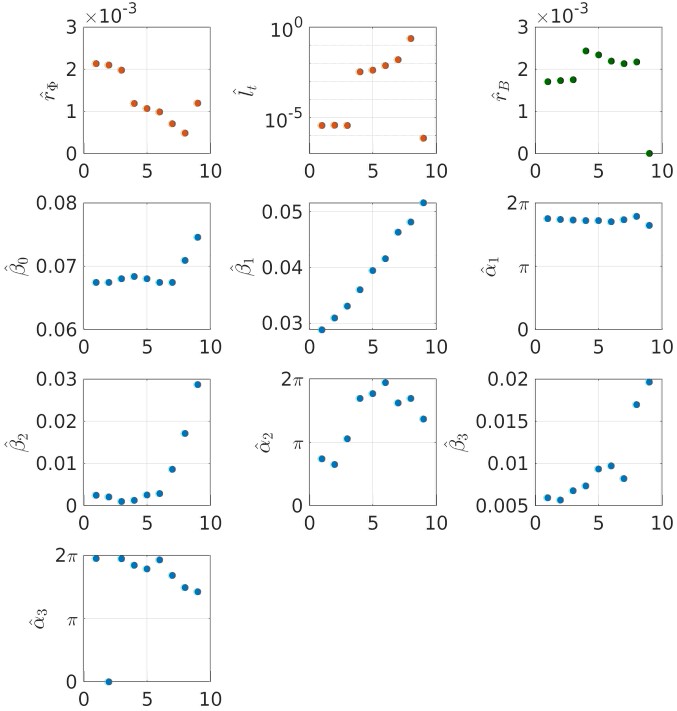

**Figure 10.** Comparison of the estimated parameters for 9 geographic area. Parameters associated to the covariance of the quantity of interest are represented in red (•); parameters associated to the mean of the quantity of interest are represented in blue (•) and the noise variance is represented in green (•).

correlation time (*e.g.,* a different *prior* distribution for $l_t$). In particular, we note that the order of magnitude of the $l_t$ values associated with this local minimum is similar to the $l_t$ values obtained for zones 4 to 7. When we reduce the geographic area, the set of observations tends to be more homogeneous, which plays in favor of models with higher correlation times.

The result obtained for zone 9 has a different interpretation. The estimated correlation time is $\hat{l}_t \simeq 7.13 \times 10^{-7}$, which is several orders of magnitude smaller than every other geographic areas. This is in conjunction with a particularly low estimated noise variance $\hat{r}_B \ll \hat{r}_\Phi$ (*cf.* Fig. 10 (c)). For this geographic area, the estimated $\hat{r}_B$ sticks to the minimum bound of the *prior* density so the optimization procedure is stuck for this parameter. Thus, the MAP estimator compensates for the remaining signal variance by increasing $\hat{r}_\Phi$ and decreasing the associated $\hat{l}_t$, so that $\mathcal{R}_\Phi$ tends towards a Dirac function, *i.e.,* a covariance

associated to white noise. In other words, the estimator is no longer able to separate the quantity of interest $\Phi$ from the noise with this dataset. To account for computational burden and stability of the estimated model we will pursue our analysis with the dataset corresponding to the 6-th geographic area.



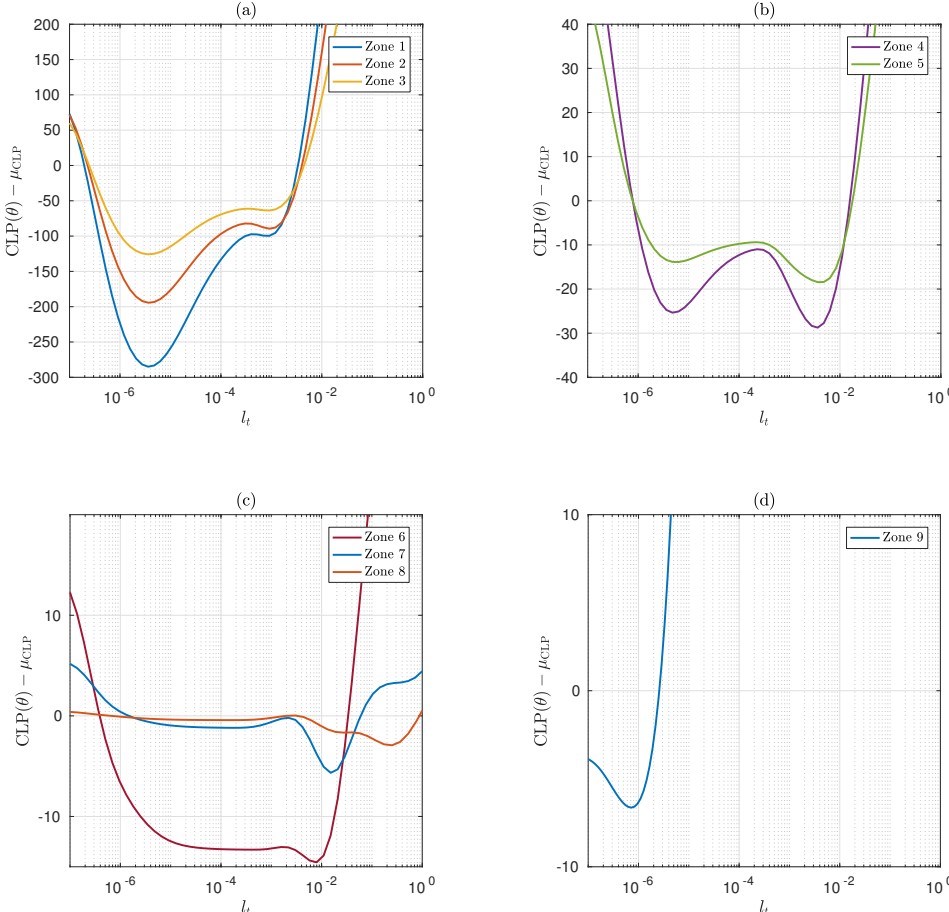

**Figure 11.** ($\mathrm{CLP} - \mu_{\mathrm{CLP}}$) as a function of the correlation time $l_t$ around $\hat{\boldsymbol{\theta}}$ for the geographic zones 1 to 3 (a), 3 to 4 (b), 6 to 8 (c) and the 9-th (d). The results for the 9 geographic areas have been grouped in 4 sketches according to their similarities.

## 6.3 Discussion on the 1-dimensional model

To conclude, it seems more interesting to carry on with geographic areas associated with higher correlation time, especially those obtained for zones 4 to 7 which corresponds to $l_t \sim 1 - 6$ days, compared to $l_t \sim 2$ minutes for zones 1 to 3. Considering the exponential covariance model reaches 5% of its total variance at $\delta_t \sim 3l_t$ (Banerjee et al., 2014), the latter tends towards its mean $m_\Phi$ after $\sim 6$ minutes. Note that the models corresponding to zones 4 to 7 are associated with variances $r_\Phi$ lower than those obtained for the zones 1 to 3 and a greater noise variance $r_B$. However, the correlation time corresponding to zones 1 to 3 is very small that the corresponding covariance tends toward a diagonal matrix (*i.e.,* a Dirac function).





### 6.4 Results in interpolation

We start with the interpolation of the LWC over the 9-year period. Note that, despite the regular sampling rate of the CPR, actual observations of altostratus in the European area are sparsely distributed. The temporal positions $t_0$ at which kriging is applied are defined by $t_0 = [0, 1/N_0, \cdots, 9 - 1/N_0, 9]$, with $N_0 = 4001$. The result of this kriging interpolation is represented in Fig. 12. The variability of the estimated $\hat{\varphi}_0$ are smaller than the actual variability of the observations, which is consistent with the estimated model parameters. Indeed, the estimated noise variance $\hat{r}_B = 2.2 \times 10^{-3}$ g$^2$.cm$^{-6}$ is greater than the estimated variance of the quantity of interest $\hat{r}_\Phi = 9.8 \times 10^{-4}$ g$^2$.cm$^{-6}$. The periodic nature of the estimate $\hat{\varphi}_0$ is strong, especially when there are no observations (for example during the 5-th year of operation of the satellite). In this case, the result of the kriging interpolation consists in the estimated mean $\hat{m}_\Phi$. In the vicinity of observations, the kriging estimates take a more complex form because of the additional information brought by the neighbouring data. The behavior observed over this period of 9 years is not surprising in itself because of the relative weakness of the observed correlation time, which is only $\hat{l}_t = 2.7$ days. We perform another kriging interpolation over a period of $\sim 36$ days centered on the beginning of the satellite's 3-rd year of operation (*cf.* Fig. 13(a)). In this figure, we observe that some observations are very closed temporally and have significantly different values of LWC which can explain the high estimated variance of the noise with respect to the variance of the quantity of interest. In this case, the kriging gives an average value of the observations. These observations generally correspond to successive profiles in the $1/50$ database, which are separated by 8 seconds. It should be noted that between two successive observations, the satellite moves $\sim 54$ km which can certainly explain a loss of spatial correlation not taken into account in our model. It is therefore likely that the model will explain this loss of spatial correlation by noise. On the other hand, we notice the presence of peaks in the structure of $\hat{\varphi}$ that can be explained by the structure of the chosen temporal covariance, which is in fact characterized by an exponential decay corresponding to the decay observed on the curve of $\hat{\varphi}$ when moving away from an observation (*cf.* Fig. 13 (a)).

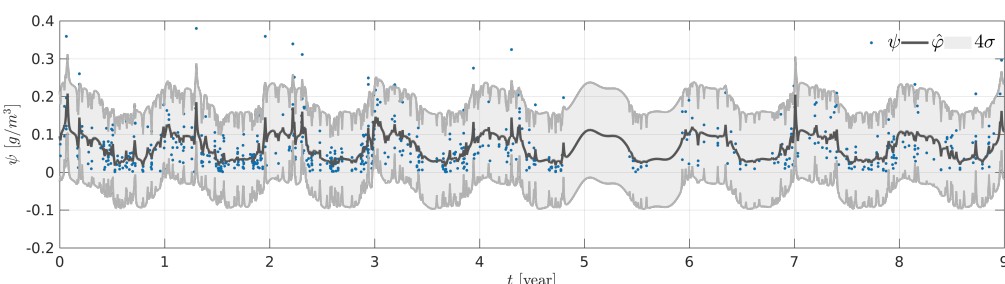

**Figure 12.** Kriging based interpolation over a 9 year period. The training set observations are represented as (•), the kriging estimate as (−), and variance of the estimation error (shaded gray).

On all kriging results, we represent the $4\sigma$ area, where $\sigma^2 = \text{var}\left[\hat{\Phi}_0 - \Phi_0\right]$ is the variance of the estimation error of (18). This variance is decomposed into two terms, the variance $\text{var}\left[\hat{\Phi}_0\right]$ and a quadratic term in $\hat{r}_{\Psi\Phi_0}(= \hat{r}_{\Phi\Phi_0})$, which represents the covariance between the object of interest and the observations. At observation location (*i.e.,* $\Phi_0$ is chosen collocated with an




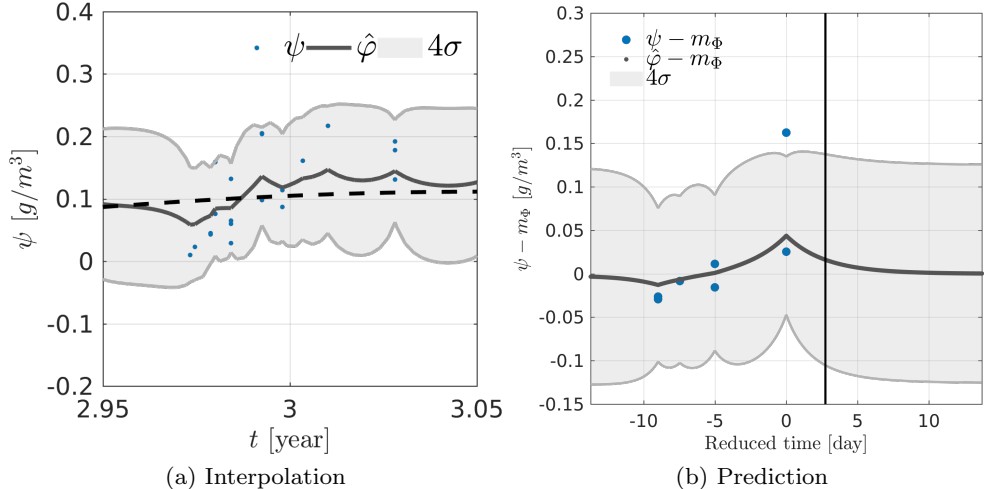

(a) Interpolation       (b) Prediction

**Figure 13.** (a) Kriging based interpolation over a period of 36 days centered at the beginning of 3-rd in the dataset. The mean is represented in as a dashed black line ($--$). (b) Kriging based prediction over a period of $\sim 36$ days centered at the beginning of 3-rd in the dataset. The vertical black line ($|$) represents the correlation time $l_t$ after the last observation. In (a) and (b), the training set observations are represented as ($\bullet$), the kriging estimate as black line ($-$), and the variance of the estimation error in shaded gray.

observation), $\hat{r}_{\boldsymbol{\Psi}\Phi_0}$ reaches its maximum value and so does the quadratic term $\hat{r}_{\boldsymbol{\Psi}\Phi_0}\hat{\boldsymbol{R}}_{\boldsymbol{\Psi}}^{-1}\hat{r}_{\boldsymbol{\Psi}\Phi_0}$. Consequently, at observation

350    location the minimum mean square error given in (18) is minimal.

### 6.5 Results in prediction

In order to study the result of kriging prediction, we selected a period that includes observations until a certain date and analyzes the behavior of the prediction beyond the last available observation (*cf.* Fig. 13(b)). In order to facilitate the interpretation, we have subtracted the estimated mean $\hat{m}_\Phi$ from the observations $\boldsymbol{\psi}$ and the estimates $\hat{\varphi}_0$ and we have centered the $x$-axis on

355    the last observation available. This clearly shows a decrease in the kriging result after the last observation towards 0, which indicates that the result tends towards the estimated mean. We plot the value of the correlation time $l_t$ as an indication. In the case of the exponential covariance, we get $5\%$ of the variance $r_\Phi$ for a time $\sim 3l_t$ which corresponds to the time from which the kriging estimation tends towards the estimated mean. It is therefore a result consistent with the chosen covariance model and the estimated correlation time. This means that the prediction will be different from the average when an observation is

360    available in a period of approximately $8.2$ days (because $l_t \simeq 2.7$ days) before or after the position where the estimation is performed.





## 6.6 Error analysis

In this section, we interpret in more details the kriging results through the analysis of the variance of the estimation error $\mathrm{var}\!\left[\hat{\Phi}_0 - \Phi_0\right]$ and the estimated noise. This is accomplished with a kriging estimation at temporal positions $t_0$ collocated to the positions of the observed data (*cf.* Fig. 14 (a)). Fig. 14 (b) represents the corresponding histograms for quantities $(\hat{\varphi}_0 - \hat{m}_\Phi)$ and $(\psi - \hat{m}_\Phi)$. Although the model used to describe the observed data is Gaussian, the histogram of the zero-mean quantity, $(\hat{\varphi}_0 - \hat{m}_\Phi)$, is not exactly Gaussian. It is actually skewed towards zero. However, its range of variation is smaller than the corresponding histogram for the observations, $(\psi - \hat{m}_\Phi)$, which is an expected result as some of the observed variability is attributed to the presence of noise.

The variance of the estimation error $\mathrm{var}\!\left[\hat{\Phi}_0 - \Phi_0\right]$ gives an approximation of the variability around the estimated value $\hat{\Phi}_0$. Note that it is impossible to compute exactly the difference $(\hat{\Phi}_0 - \Phi_0)$, but we have estimated the variance of the noise knowing the observations, it is then possible to compute the variance $\mathrm{var}\!\left[\hat{\Phi}_0 - \Phi_0\right]$ (*cf.* Sect. 3). This last variance term can be used to compute *posterior* realizations of the quantity of interest $\Phi$. Fig. 14 (c) represents the variance $\mathrm{var}\!\left[\Phi_0\right]$ and the variance of the estimation error $\mathrm{var}\!\left[\hat{\Phi}_0 - \Phi_0\right]$. The latter has a lower magnitude than $\mathrm{var}\!\left[\Phi\right]_0$ which is consistent with (11) as the term $r_{\Psi\Phi_0}^{\mathrm{t}}\boldsymbol{R}^{-1}r_{\Psi\Phi_0}^{\mathrm{t}}$ is positive.

Finally, we computed the difference $(\psi - \hat{\varphi}_0)$ (*cf.* Fig. 15(a)). According to (2), this distribution must correspond to the noise distribution. We represented the normalized histogram of $(\psi - \hat{\varphi}_0)$ in Fig. 15 (b) as well as the pdf of the noise $B \sim \mathcal{N}(\mathbf{0}, \hat{r}_B)$. We observe that the two distributions have slightly different shapes but similar ranges of variation. The differences observed are mainly due to the fact that the parameters of a Gaussian model are estimated from observations that do not have a strictly Gaussian distribution. Indeed, we notice that the observations have an asymmetric distribution which extends towards higher values of LWC than the Gaussian distribution.

## 7 The 2-dimensional time-altitude case

In a similar fashion as Sect. 6, we examine the 2-dimensional time-altitude case starting from the parameter estimation followed by the kriging results.

## 7.1 Parameter estimation

We are considering here the estimation of model parameters including the altitude dimension. Following the conclusions of the previous section, we use the observations of the 6-th geographic area. However, for computational reasons, the training dataset is restricted to the first 3 years of observations, which consists of $N_{\mathrm{obs}} = 4167$ observations (*cf.* Fig. 17).

Fig. 16 represents the evolution of each parameter during the optimization process. We observe that all parameters have stabilized towards an estimated value. Moreover, none of the parameters have converged towards the bound defined by the prior.





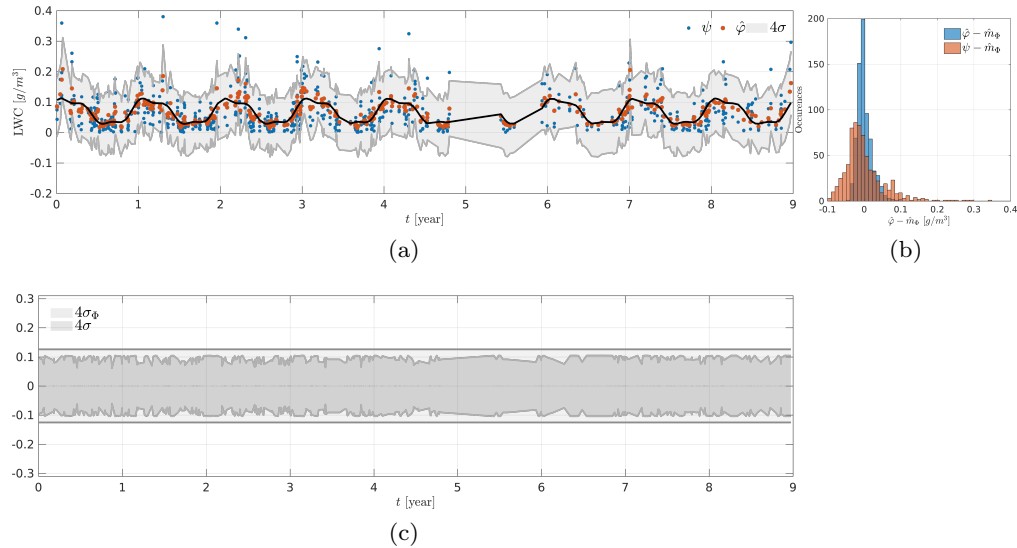

**Figure 14.** Kriging results collocated to the observations. (a) Observations $\psi$ (•), estimate $\hat{\varphi}_0$ (•), estimated mean $\hat{m}_\Phi$ (—) and variance of the estimation error in shaded gray. (b) Histogramms of $(\hat{\varphi}_0 - \hat{m}_\Phi)$ and $(\psi - \hat{m}_\Phi)$. (c) Shaded gray representation of the variance of the estimation error and the variance of the quantity of interest $\Phi$.

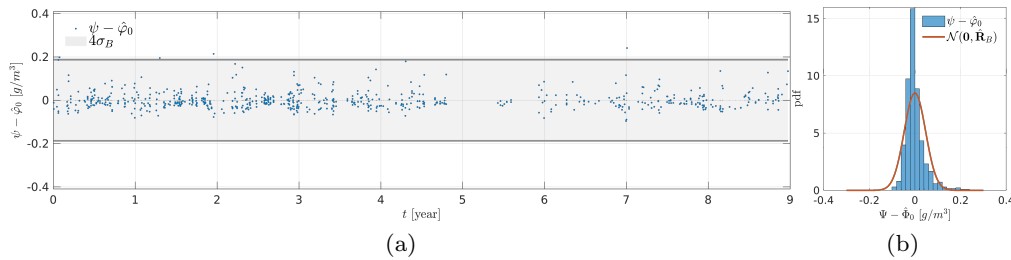

**Figure 15.** (a) Differences $(\psi - \hat{\varphi}_0)$(•) superimposed on $4\sigma_B$ in shaded gray, and (b) normalized histograms of $(\psi - \hat{\varphi}_0)$ and the pdf of $B \sim \mathcal{N}(\mathbf{0}, \hat{\mathbf{R}}_B)$.

The estimated values in the 1-dimensional (temporal) and 2-dimensional (time-altitude) cases are consistent from one model to the other (*cf.* Table 2). Overall, the order of magnitude for most quantities are close. However, we point out some noticeable differences:

395      – there is a factor 10 between the estimated variance $\hat{r}_\Phi$ of the time model and the time-altitude model;

      – for the 1-dimensional case, $\hat{r}_\Phi < \hat{r}_B$ whereas in the 2-dimensional case we have $\hat{r}_\Phi > \hat{r}_B$, which means adding a dimension to the model helps to find some structure in the data that were missing otherwise;

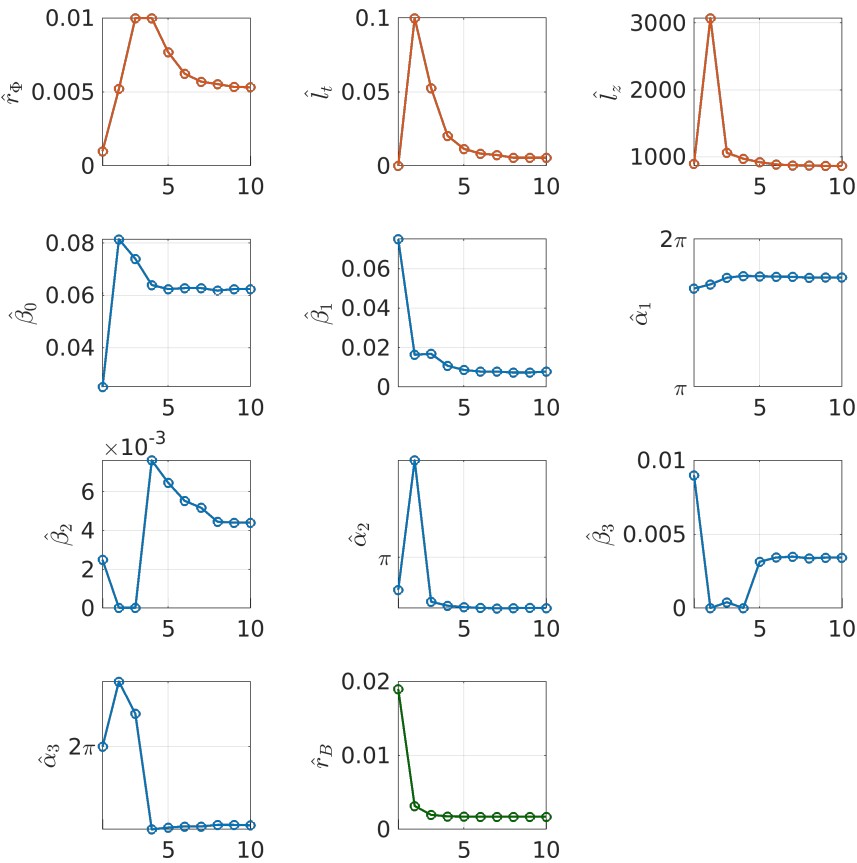

**Figure 16.** Evolution of each parameters during the optimization process in 2-dimensional case (time-altitude). Parameters associated to the covariance of the quantity of interest are represented in red (○); parameters associated to the mean of the quantity of interest are represented in blue (○) and the noise variance is represented in green (○).

– the amplitudes and phases of the first and second harmonics ($\hat{\beta}_2$, $\hat{\beta}_3$, $\hat{\alpha}_2$ and $\hat{\alpha}_3$) differ substantially.

This last point explains the distinct mean behavior for the two models. The mean of altitude-time model has a lower amplitude which can be explained by the fact that the model must accommodate for a greater variability of observations due to an underlying altitude dependence of the dataset. It is an indication that future development should include this altitude dependence.

## 7.2 Kriging results

In this section, we present the kriging results obtained for the time-altitude model. The parameters corresponding to this model have been estimated in Sect. 7.1. The kriging estimator is applied on a regular grid of $20 \times 500$ positions in time and altitude, this corresponds to a total of 10000 positions. We take $z_0 = [1, \cdots, 8]$ km and $t_0 = [2, \cdots, 5.5]$ years. In addition, we excluded observations made after the 5-th year. Therefore, we used a set of $N_{\text{obs}} = 6112$ observations. Fig. 17(a) outlines the situation we have just defined. The obtained kriging surface represented in Fig. 17(b) visually fits the observations from Fig. 17. Fig. 17(c)





**Table 2.** Comparison between estimated parameters for the 1d and the 2d model. Note that we have fixed $T = 1$ in the optimization procedure.

| $\hat{\boldsymbol{\theta}}$ | 1-D Model | 2-D Model |
|---|---|---|
| $\hat{r}_\Phi$ | $\sim 9.84 \times 10^{-4}$ | $\sim 5.3 \times 10^{-3}$ |
| $\hat{l}_t$ | $\sim 7.5 \times 10^{-3}$ | $\sim 5.6 \times 10^{-3}$ |
| $\hat{l}_z$ | / | $\sim 865$ |
| $\hat{r}_B$ | $\sim 2.2 \times 10^{-3}$ | $\sim 1.7 \times 10^{-3}$ |
| $\hat{\beta}_0$ | $\sim 6.7 \times 10^{-2}$ | $\sim 6.2 \times 10^{-2}$ |
| $\hat{\beta}_1$ | $\sim 4.2 \times 10^{-2}$ | $\sim 7.9 \times 10^{-3}$ |
| $\hat{\beta}_2$ | $\sim 2.9 \times 10^{-3}$ | $\sim 4.4 \times 10^{-3}$ |
| $\hat{\beta}_3$ | $\sim 9.7 \times 10^{-3}$ | $\sim 3.4 \times 10^{-3}$ |
| $\hat{\alpha}_1$ | $\sim 5.4$ | $\sim 5.5$ |
| $\hat{\alpha}_2$ | $\sim 6.1$ | $\sim 1.5$ |
| $\hat{\alpha}_3$ | $\sim 6.1$ | $\sim 3.6$ |

represents the 2-dimensional map of the variance of the estimation error. We note that the variance of the estimation error is minimal when there are observations in the neighborhood, whereas it increases as we move away from the observations. This is an expected result that is consistent with what has been observed in the case of time kriging. Since the interpretation of kriging surfaces is complex, we represent these results at constant altitude (*cf.* Fig. 18). On constant height sections, the estimation structure seems more complex than in the 1-dimensional case, it can be explained by the interaction with observations at altitudes above and below the considered altitude. Moreover, we note that the model tends to the mean of the model when we move away from observations. This is especially true in the case of long-term prediction around the 5-th year.

# 8   Conclusion and perspectives

The interpolation, in space, and prediction, in time, of the cloud microphysics in medium and long term are of major importance in weather and climate analysis. Since a perfect estimation is obviously unattainable, it is an issue of uncertainty quantification. In this original work, we develop a statistical spatio-temporal kriging-based approach that is able to interpolate/predict from the dataset and provide uncertainties. Beforehand, it requires in particular estimating the covariance model parameters; it is performed in a Bayesian setting, which allows for estimation and uncertainty quantification. The approach is then applied to a subset of the CloudSat dataset which shows promising results, especially in the 2-dimensional case where detailed structure appears in the quantity of interest. A natural extension to this approach would be to consider the latitude and longitude variables in order to interpolate horizontally the quantity of interest. Another possible extension would be to consider the remaining cloud types (cumulus, stratus, *etc*...). Finally, the impact of the parameter uncertainty in the kriging results could be rigorously handle by developing a complete Bayesian hierarchical model.





(a)

(b)

(c)

**Figure 17.** CloudSat observations on geographic area 6. The red rectangle (□) indicates the training dataset, the green rectangle (□) indicates observations not used in the kriging estimate and the plain black lines represent the spatialtemporal positions where the kriging estimates are computed (a). Result of the 2-dimensional kriging on zone 6 (b) and its associated 2-dimensional variance of the estimation error (c).





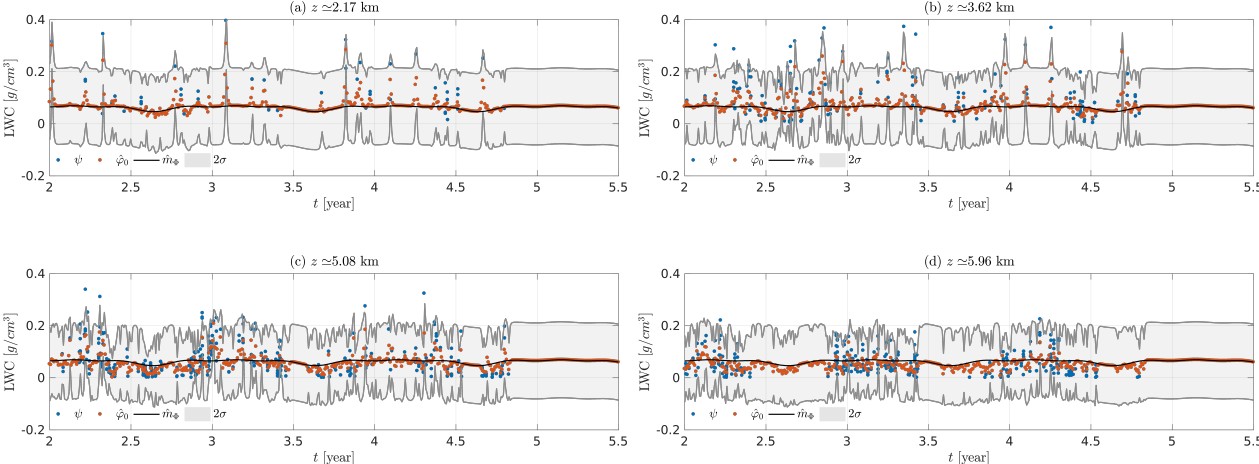

**Figure 18.** Kriging results at constant altitudes, $z \simeq 2.17$ km (a), $z \simeq 3.62$ km (b), $z \simeq 5.08$ km (c), $z \simeq 5.96$ km (d). We represent the observations $\psi$ ($\bullet$), the estimations $\hat{\varphi}_0$ ($\bullet$), the mean ($-$) as well as $2\sigma$ around the estimation $\hat{\varphi}_0$.

*Data availability.* The level 2B-CWC-RO CloudSat data used in this study have been downloaded from the CloudSat Data Processing Center (http://www.cloudsat.cira.colostate.edu/).

*Author contributions.* PM and JFG designed the preliminary study, with continuous discussions from all co-authors along the project. JML developed the model code and performed the simulations. JML prepared the manuscript with contributions from all co-authors.

*Competing interests.* The authors declare that they have no conflict of interest.

*Acknowledgements.* The authors would like to thank Audrey Giremus for her contribution to the preliminary step of the work.



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
