# Peer review of "A kriging-based analysis of cloud Liquid Water Content using CloudSat data"

_Atmospheric Measurement Techniques, 2021_

## Author Comment (AC1)

**A kriging-based analysis of cloud Liquid Water Content using CloudSat data**

Atmospheric Measurement Techniques

March 4, 2022

J.-M. Lalande[1,*], G. Bourmaud[1],
P. Minvielle[2], J.-F. Giovannelli[1]

jean-marie.lalande@meteo.fr

[1]IMS (Univ. Bordeaux, CNRS, Bordeaux INP), 33400 Talence, France
[2]CESTA, DAM, CEA, 33114 Le Barp, France
[*]Now at CNRM, Université de Toulouse, Météo-France, CNRS, Lannion, France

**Contents**

**Abstract**

We thank the two anonymous reviewers for their efforts and feedback.

**1 Reviewer 1**

This article proposed a statistical spatio-temporal kriging-based approach that is able to interpolate/predict from the dataset and provide uncertainties. The topic is of interest to the readership of this journal, this study is well planned, and the mathematics appears correct and keeps at an appropriate level. However, this paper still need to be improved with moderate revisions:

> **Reviewer 1 Comment 1**
>
> Kriging methods have been widely used in spatial and temporal interpolation for meteorological elements, in the meantime, many improved kriging methods have been proposed, such as Universal Kriging, Co-Kriging, Disjunctive Kriging and so on. Therefore, this paper should give a brief overview of these improved methods, and highlight the advance of this paper method compared with the existing methods.

**Response**

We agree with the reviewer that the manuscript would greatly benefit from a better overview of kriging methods applied to meteorological parameters, as well as more references about the kriging literature in general. In that sense, we propose to add the following paragraph in the manuscript (in the introduction) in order to address this comment:

> A number of statistical methods dedicated to the analysis of spatial and spatio-temporal data have been developed over the years taking into account the spatial and/or temporal correlation of the observations Ripley (1981); Cressie (1993). Among them, the kriging estimator was initially introduced by Krige (1951), from which it takes its name, to estimate the gold distribution at the Witwatersrand reef complex in South Africa based on samples from boreholes. It was then formalized mathematically by Matheron (1963) in the context of mining geology. Afterwards, the kriging estimator spread to many other areas of sciences Wackernagel (2013) (hydrogeology, geotechnics, agronomy, air quality, fishery, epidemiology, water and soil pollution, noise, *etc.*). Best-known kriging techniques are Simple Kriging, which assumes stationarity of the $1^{st}$-order with a known mean and Ordinary Kriging, where the mean is unknown. Since its first development, kriging techniques have largely evolved, and a number of new kriging techniques have been developed Chiles and Delfiner (1999); Cressie (1993); Cressie and Wikle (2015). In the field of meteorology, the kriging estimator have mostly been used in order to estimate precipitation accumulation from rain gauges Nour et al. (2006); Belo-Pereira et al. (2011) and in combination with satellite-derived precipitation Jewell and Gaussiat (2015); Verdin et al. (2016); Varouchakis et al. (2021) as well as the estimation of aerosols concentration in the air from in-situ observations Park (2016). It has also been used for the estimation of temperature from in-situ measurements Heuvelink et al. (2012); Didari and Zand-Parsa (2018), from satellite observations Florio et al. (2004) or a combination of them Didari and Zand-Parsa (2018), and for the estimation of surface properties from remote measurements der Meer (2012); Zakeri and Mariethoz (2021).

The introduction has been slightly reorganized to include this paragraph. Morevover, as the reviewer stated, some of the extended kriging estimators include:

1. the Universal Kriging assumes a non-stationary random function at the 1st-order, but $2^{nd}$ order stationary, hence the mean model is written such as $\boldsymbol{\mu}(\mathbf{s}) = \boldsymbol{\beta}^t \mathbf{f}(\mathbf{s})$ with parameters acting linearly with a set of linear and polynomial function such as $\mathbf{f}(\mathbf{s}) = [\mathbf{1}, \mathbf{s}, \mathbf{s}^2, \cdots]$ in the model. In that sense, our estimator is similar to the Universal Kriging estimator as our mean is a linear combination of cosinus functions. However, in order to estimate the phase parameters of the cosinus, which are non-linears, we had to modify the estimation approach used in Universal Kriging. Our contribution, then, rest upon a mean adapted to our problem that includes the periodicity of the observations.

2. Co-Kriging which is applied to the estimation of multivariate random variable that are supposedly statistically correlated. In our case we aim at estimating a single variables (Liquid Water Content) from spatio-temporal observations so that we don't need to resort to this approach. Future development could include the joint estimation of LWC/IWC. Co-kriging is an extension of ordinary kriging. Furthermore, co-kriging can become computationally intensive to use as it requires to estimate the covariance

for each variables as well as their cross-covariance, this becomes rapidly difficult when dealing with big amount of data (such as satellite observations).

3. Disjunctive Kriging is a kriging techniques that has been available for over 50 years and falls in the field of nonlinear geostatistics. This methods is dedicated to the estimation of some functions of the quantity of interest $f[\phi(x_0)]$ instead of $\phi(x_0)$. In that case considering linear combination of the observation is not sufficient and nonlinear kriging methods have been proposed. Nonlinear kriging was designed to face the more and more complicated mining evaluation problems, specifically, instead of estimating the proportion of ore in a block, the problem was to estimate if the block in question would exceed a threshold between ore and waste Rivoirard (1994). Disjunctive Kriging arises as the co-kriging of indicator functions that are used to express any function $f$ of the quantity of interest. The original problem does not require to resort to such approach but we note that this estimator should be considered to solve other interesting scientific questions (*i.e.* estimate if a specific cloud is going to exceed some LWC threshold, classification problems, *etc.*)

A number of other kriging estimator has been developed over the years so that it is nearly impossible to give an exhaustive listing. Our kriging estimator is an adaptation of the Universal Kriging estimator to include a more complex mean for the quantity of interest. The two other kriging estimators cited by the reviewer (co-kriging and disjunctive kriging), while they could be investigated in our context (*i.e.,* CloudSat observations), serve different purposes and are not specifically adapted to our problem. However, in our perspective, we mention some other estimation problems (*i.e.,* joint estimation IWC/LWC, integrate different cloud types) for which these estimators should be considered (we slightly modified the Conlusion and Perspective part of the manuscript in that sense). Reviewing the literature, when we first got our hands on this study, we found out confusing the difference in treatment of the model parameters associated with the mean and those associated with the covariance. One of the starting point of our analysis, along with the specific application to the CloudSat observations, was to specify a unified treatment of the mean and the covariance parameters and then to estimate these parameters, with the MAP estimator, before applying the kriging equation. We believe such an approach gives a better estimation of the variance of the estimation error.

> **Reviewer 1 Comment 2**
>
> The experimental section is lack of adequate contrast experiments with other existing interpolation methods (especially the representative improved kriging methods).

**Response**

Our study was initially motivated to demonstrate the feasibility of the kriging estimator in the complex situation of the estimation of cloud liquid water content at spatial location not sampled by Cloudsat observations and to derive a generative model of the LWC distribution, including the uncertainties associated with estimated quantity of interest. Moreover, we also wanted to give an in-depth evaluation of the kriging estimator in order for any interested reader to be able to reproduce this specific experiment. A number of other studies have investigated the performance of kriging estimator in comparison with other interpolation/extrapolation methods Lam (1983); Caruso and Quarta (1998); Stein (1999), we believe

such comparison doesn't fall in the scope of this particular paper and would considerably lengthen the study at the cost of clarity.

Moreover, our philosophy was to go back to the basics in a context never treated before and designed a new version of the kriging estimator for our purposes. One of our main concern, in that sense, was to increase the interpretability of the model parameters by carefully choosing them from the exploratory analysis. The detailed discussion on the estimated parameters could hardly be extended to other interpolation or kriging methods as the model parameters would definitely be different and hardly comparable. Our opinion is that this should preferably be done in a different study.

**2    Reviewer 2**

In this article, the authors use a kriging method to interpolate measurements and predict cloud properties of LWC from CloudSat satellite retrievals. Because of the polar orbit and the fixed-nadir radar measurement, CloudSat provides the most spatiotemporally coarse measurements of any satellite cloud radar. This is a frequent impediment for users of CloudSat for climatologies, and many prior studies have dealt with this issue with comparatively simple methods, such as gaussian means and standard deviations. There is a clear benefit in an advanced statistical method that could provide well-described (that is, with numerical uncertainties) predictions of unsampled regions based off of neighboring retrievals, so I recommend this paper for publication.

The majority of this paper discusses the kriging method, application, and resulting optimal estimation. I do not have much of a background in Kriging, but it seems like the other reviewer has already gone in-depth with recommendations on this matter, so I will not add anything else. I do not find any other issues with the results.

**Response**

We thank the reviewer for his laudatory comments. As being asked, we refer him to the answer to the first reviewer.

**References**

Belo-Pereira, M., Dutra, E., and Viterbo, P. (2011). Evaluation of global precipitation data sets over the iberian peninsula. *Journal of Geophysical Research: Atmospheres*, 116(D20).

Caruso, C. and Quarta, F. (1998). Interpolation methods comparison. *Computers and Mathematics with Applications*, 35(12):109–126.

Chiles, J.-P. and Delfiner, P. (1999). *Geostatistics: modeling spatial uncertainty*. Wiley, New York.

Cressie, N. (1993). *Statistics for Spatial Data*. Wiley Series in Probability and Statistics. Wiley.

Cressie, N. and Wikle, C. (2015). *Statistics for Spatio-Temporal Data*. Wiley.

der Meer, F. V. (2012). Remote-sensing image analysis and geostatistics. *International Journal of Remote Sensing*, 33(18):5644–5676.

Didari, S. and Zand-Parsa, S. (2018). Enhancing estimation accuracy of daily maximum, minimum, and mean air temperature using spatio-temporal ground-based and remote-sensing data in southern iran. *International Journal of Remote Sensing*, 39(19):6316–6339.

Florio, E. N., Lele, S. R., Chang, Y. C., Sterner, R., and Glass, G. E. (2004). Integrating avhrr satellite data and noaa ground observations to predict surface air temperature: a statistical approach. *International Journal of Remote Sensing*, 25(15):2979–2994.

Heuvelink, G., Griffith, D., Hengl, T., and Melles, S. (2012). Sampling design optimization for space-time kriging. *Spatio-Temporal Design: Advances in Efficient Data Acquisition*, pages 207–230.

Jewell, S. and Gaussiat, N. (2015). An assessment of kriging based rain-gauge-radar merging techniques. *Quarterly Journal of the Royal Meteorological Society*, 141.

Krige, D. (1951). A statistical approach to some basic mine valuation problems on the witwatersrand. *Journal of the Southern African Institute of Mining and Metallurgy*, 52(6):119–139.

Lam, N. (1983). Spatial interpolation methods: a review. *American Cartographer*, 10:129–149.

Matheron, G. (1963). Principles of geostatistics. *Economic Geology*, 58(8):1246–1266.

Nour, M., Smit, D., and Gamal El-Din, M. (2006). Geostatistical mapping of precipitation: implications for rain gauge network design. *Water Science and Technology*, 53(10):101–110.

Park, N.-W. (2016). Time-series mapping of pm10 concentration using multi-gaussian space-time kriging: A case study in the seoul metropolitan area, korea. *Advances in Meteorology*, 2016:1–10.

Ripley, B. D. (1981). *Spatial statistics*. Wiley New York.

Rivoirard, J. (1994). *Introduction to Disjunctive Kriging and Non-linear Geostatistics*. Spatial information systems. Clarendon Press.

Stein, M. L. (1999). *Interpolation of spatial data*. Springer Series in Statistics. Springer-Verlag, New York. Some theory for Kriging.

Varouchakis, E. A., Kamińska-Chuchmala, A., Kowalik, G., Spanoudaki, K., and Graña, M. (2021). Combining geostatistics and remote sensing data to improve spatiotemporal analysis of precipitation. *Sensors*, 21(9).

Verdin, A., Funk, C., Rajagopalan, B., and Kleiber, W. (2016). Kriging and local polynomial methods for blending satellite-derived and gauge precipitation estimates to support hydrologic early warning systems. *IEEE Transactions on Geoscience and Remote Sensing*, 54(5):2552–2562.

Wackernagel, H. (2013). *Multivariate Geostatistics: An Introduction with Applications*. Springer Berlin Heidelberg.

Zakeri, F. and Mariethoz, G. (2021). A review of geostatistical simulation models applied to satellite remote sensing: Methods and applications. *Remote Sensing of Environment*, 259:112381.